# Ultra-Wideband Common-Mode Rejection Structure with Autonomous Phase Balancing for Ultra-High-Speed Digital Transmission

**DOI:** 10.3390/s24196180

**Published:** 2024-09-24

**Authors:** Byung-Cheol Min, Jeong-Sik Choi, Hyun-Chul Choi, Kang-Wook Kim

**Affiliations:** School of Electronic and Electrical Engineering, Kyungpook National University, Daegu 41566, Republic of Korea; minbc4658@knu.ac.kr (B.-C.M.); jeongsik2@knu.ac.kr (J.-S.C.); hcchoi@ee.knu.ac.kr (H.-C.C.)

**Keywords:** balanced line, ultra-wideband, ultra-high-speed digital transmission, common-mode rejection filter, phase balancing

## Abstract

For ultra-high-speed digital transmission, required by 5G/6G communications, ultra-wideband common-mode rejection (CMR) structures with autonomous phase-balancing capability are proposed. Common-mode noise, caused by phase and amplitude unbalances, is one of the most undesired disturbances affecting modern digital circuits. According to the circuit design guides with a typically used differential line (DL) for high-speed digital transmission, common-mode rejection is achieved using CMR filters, and the unbalanced phase, caused by a length difference between the two signal lines of a DL, is compensated by inserting an additional delay line. However, due to nonlinear phase interactions between the two DLs and unbalanced electromagnetic (EM) interferences, the conventional compensation method is frequency-limited at around 10 GHz. To significantly enhance the common-mode rejection level and extend the phase recovery bandwidth, the proposed CMR structure utilizes a planar balanced line (BL), such as a coplanar stripline (CPS) or a parallel stripline (PSL), along with additional conductor strips arranged laterally near the BL. To demonstrate the performance of the proposed BL-based CMR structures, various types of CMR structures are fabricated, and the measurement results are compared with the 3D EM simulation results. As a result, it is proven that the proposed BL-based CMR structures have the capability to reject the common-mode noise with suppression levels of more than 10 dB and to simultaneously recover the phase balance from near DC to over 40 GHz.

## 1. Introduction

Due to the significant increase in demand for massive digital data transmission for 5G/6G wireless communications and artificial intelligence, a digital circuit has been encountered with the challenge of ultra-high-speed data rates over 100 Gbps. In modern digital circuits, the digital data rate often reaches 10 s of Gbps, and a high-speed digital signal, composed of the fundamental frequency and its multiple harmonics, tends to occupy a very wide frequency bandwidth, extending up to mm wave frequencies. The spanned spectrum of ultra-high-speed digital signals is often significantly wider than the bandwidth of the analog part of the system, such as the RF front-end modules, and surpasses even the bandwidth of ultra-wideband (UWB) RF systems. In high-speed digital circuits, many undesired phenomena related to the microwave/mm wave frequency components of the digital signals may occur.

For the high-speed digital transmission, differential signaling is typically adopted with the use of a differential line (DL) as a digital transmission line, which consists of two parallel microstrip lines with a gap. Differential signaling, with an oppositely polarized signal on each line, enables it to effectively resist external noises and to operate at lower signal voltages, and the DL has been a preferred transmission line instead of a single-ended line, especially for high-speed digital transmission. For an ideal DL, the two signals of the DL with the opposite polarities are independent of each other and have an equal amplitude with the opposite phases, and the sum of them would be zero.

However, in practical circuit environments, the two signals can be unbalanced in amplitude and/or phase, and the sum of the two signals may produce an unwanted signal component called common-mode noise, which is a cause of many undesirable phenomena [1,2,3]. The common-mode noise degrades the quality of the differential signal and becomes a source of electromagnetic (EM) interference, adversely affecting nearby lines and circuits. Also, it may cause short-term noise in the chip or element and long-term damage to the receiver. Furthermore, common-mode noise may generate a common-mode current in the ground plane, which can flow to cables and connectors, resulting in common-mode radiation known as radio frequency interference (RFI) [4,5,6]. Therefore, especially with high-speed digital transmission, the minimization of the common-mode noise is very important.

The main source of the common-mode noise is a signal skew, which is caused by phase unbalance and/or amplitude unbalance [7]. The phase unbalance is mostly generated by a length difference between the two signal lines of the DL due to curves or unbalanced pin maps of interconnectors. However, it is noteworthy that, in such cases, amplitude unbalance between the two signal lines is also accompanied [8]. The level of phase unbalance of the differential signal is an amount of phase difference deviation from 180°. To recover from the phase unbalance, conventional digital circuit design guides have recommended a compensation method of adding an extra line length for the shorter signal line [9,10,11,12,13]. This compensation can be very effective if the length difference between the two signal lines of the DL is precisely estimated, and the two DLs are electromagnetically independent. However, as the signal frequency exceeds ~10 GHz, the EM coupling between the two DLs causes a nonlinear phase interaction, adversely affecting the signal balance [14]. Furthermore, with FR-4 substrates, even when the lengths of the two signal lines are the same, a significant amount of signal skew can be generated depending on the orientation between the signal line and the fiber weave (“fiber weave effect”), which is very hard to predict and becomes serious at frequencies greater than 10 GHz [15,16]. Also, several studies have proposed methods to suppress the common-mode noise arising from the signal skew. However, most of these methods offered suppression bandwidths mostly up to 10 GHz [17,18,19,20].

The amplitude unbalance of the DL is mostly caused by uneven electromagnetic coupling on the two signal lines of the DL or by gain mismatch in the differential circuit. When the digital signal distortion is only due to the amplitude unbalance, the skewed signal can be restored by using a common-mode rejection (CMR) filter, and a variety of CMR filters have been proposed. CMR filters using defected ground structures (DGSs) were developed with common-mode suppression levels of 10 dB and 15 dB in frequency bands of 3.7–10.8 GHz and 3.2 GHz–12.4 GHz, respectively [21,22]. In addition, a CMR filter using interdigital resonators was proposed to have a suppression level of 7 dB in a frequency range of 4.6 GHz to 20 GHz [23]. In [24], a CMR filter using an absorber was introduced, which showed a suppression level of 10 dB in a frequency band of 2 GHz to 11 GHz. Additionally, a CMR filter using an electromagnetic bandgap (EBG) was developed to possess a suppression level of 10 dB in a frequency band from 3.1 GHz to 10.6 GHz [25]. Also, CMR filters using a quarter-wavelength resonator and a split-ring resonator were developed with common-mode suppression levels of 10 dB in frequency bands of 2.34 GHz–3.06 GHz and 3.53 GHz–10.1 GHz, respectively [26,27]. However, the bandwidths of the reported CMR filters were mostly limited to around 10 GHz. In order to remove the signal skew of a DL, thereby improving signal integrity, both the phase and amplitude unbalances should be restored. So far, there have been no reports of other devices or structures that can simultaneously balance the phase and amplitude of an unbalanced differential signal (a skewed digital signal) in wide or ultra-wide bandwidth.

On the other hand, ultra-wideband transmission line structures based on balanced lines (BLs) utilizing the optimized DL-to-BL transition structures were proposed by the authors’ group, providing more than 40 GHz of bandwidth to support ultra-high-speed digital data transmission [28]. The BL supports differential signaling and consists of two signal lines that are symmetrically aligned with opposite polarities. Two representative planar BLs are a coplanar stripline (CPS) and a parallel stripline (PSL), which are often used to propagate balanced signals in the microwave/mm wave circuits. The two signal lines of the BL carry opposite polarity signals, which together support a balanced signal with tight EM coupling, and each line serves as a ground line for the other.

Utilization of a BL as a digital transmission line provides numerous advantages that can help overcome the limitations of DLs [28]. First, BLs can reduce common-mode noise in ultra-wide frequency bandwidths, providing the common-mode noise rejection level of 7~14 dB from near DC to over 40 GHz. Another very important property of BLs is their ability to restore the signal balance when an incoming digital signal is distorted [14,28]. Tight EM coupling between the BLs inherently promotes the restoration of signal balance between the two BLs to have the same amplitude and the opposite-polarity phases. This autonomous phase-balancing action in a CPS-fed balanced antenna was reported in detail by the authors’ group [14], and the phase recovery phenomenon of a distorted digital signal was first reported in [28]. Since most of the current digital components and printed circuit boards (PCBs) are based on DLs, the ultra-wideband transitional structures (i.e., DL-to-BL transitions) were proposed to optimally connect a DL to BLs with more than 40 GHz bandwidth [28]. It is noted that the operation frequency bandwidth is mostly limited by the performance of the transition structures [29]. This autonomous phase recovery action, however, can be limited by the phase unbalance level of the incoming signal, i.e., when exceeding the phase unbalance of ~λ/8.

In this paper, ultra-wideband digital transmission structures that simultaneously restore the phase and amplitude of a skewed digital signal, enhancing the common-mode rejection level and the autonomous phase balance up to over 40 GHz, are proposed. By arranging extra conductor strips adjacent to the BLs, it is proven that the common-mode rejection capability as well as the phase recovery action can be significantly enhanced. The additional conductor structure acts to significantly reject the common-mode components of the skewed digital signals with minimal effects on the differential signals, also providing the autonomous phase recovery. In order to prove the enhanced performances of the proposed structures, three digital transmission structure models are considered: the DL-based structure, the BL-based structure, and the proposed BL-based structure with adjacent conductor strips. With the DL-based circuits, the process of a signal skew generation and the limitations of the compensation method are discussed. Also, with the BL-based circuits, the actions of the autonomous phase recovery and common-mode rejection are described. In addition, with the BLs combined with adjacent metal strips, the processes of enhanced common-mode rejection and phase recovery, resulting in improved signal integrity, are described. In order to prove the performances of the compensation methods related to the three models, these model structures are fabricated and measured. Through EM simulations and measurements, the proposed ultra-wideband digital transmission structure is validated to perform in ultra-wideband bandwidth with excellent phase recovery and common-mode rejection, applicable to the ultra-high-speed digital data transmission required for beyond 5G communications.

## 2. Common-Mode Noise in a Digital Circuit

A differential line (DL), supporting differential signaling with oppositely polarized signals, has been a preferred transmission line, especially for high-speed digital transmission. However, there are various undesired phenomena that can happen in a conventional digital circuit, which consists of multiple chips, interfaces, and signal lines in a limited space, as shown in Figure 1. A bent line or an unbalanced pin map causes a length difference between the two signal lines of a DL, which generates a phase unbalance. Also, crosstalk between the adjacent signal lines or a gain mismatch in the differential circuit leads to an unbalanced amplitude of the differential signal. The phase and amplitude unbalances are the major causes of common-mode noise, which degrades the quality of the differential signal and triggers EM interference and RFI, which inevitably occur due to the complexity of the digital circuit. Therefore, these days, to improve signal integrity, digital circuit designers should follow the strict design guides for the minimization of phase and amplitude unbalances on high-speed digital signal lines.

Common-mode noise can be generated in the presence of phase and/or amplitude unbalances between the two signal lines of a DL. Figure 2 shows waveform cases which generate common-mode noises: phase unbalance and amplitude unbalance. When the amplitudes of the two signals are different, common-mode noise can be generated even if the phase is balanced (180° difference between the two signals). A CMR filter can balance the signal amplitudes of the two lines within the operation frequency bandwidth. However, when a distorted digital signal is caused by both amplitude and phase unbalances, the amplitude balanced signal, passing through the CMR filter, may still remain distorted as compared with the original digital signal waveform; i.e., the deformed level depends on the amount of phase delay.

On the other hand, when a phase unbalance is generated due to a length difference between the two lines of a DL, an additional delay line is connected to the shorter line of the DL to compensate for the length difference, as recommended by the conventional design guide. It is recommended that the delay line on the FR-4 substrate be applied within 15 mm (590 mil) [30]. If the two lines of the DL are electromagnetically independent, the delay-line compensation can be very effective. However, since there is a certain level of EM coupling between the two DLs, it was reported that nonlinear amplitude and phase interactions between the two lines could occur, especially at frequencies over 10 GHz [28].

As an example of this nonlinear phase interaction, the EM-simulated phase difference between the DLs as a function of frequency is shown in Figure 3. For the EM simulations, a commercial 3D EM simulator (CST EM Suite 2022) is used. In the figure, a line length difference of 60 mil is introduced on the left side, and a length compensation section with a delay line of the DL is applied at the shorter line of the DL. The DL structure is designed on the 8 mil FR-4 substrate, and the length compensation section is applied at a distance of 590 mil from the delay location, which follows the digital circuit design rule. Ideally, if the two DLs are independent from each other, the phase difference between the two DLs after the compensation should be very close to 180°. Considering the actual PCB environments, a skew budget for high-speed digital data is recommended as 0.14 UI (unit interval), which can be converted to a phase difference of 25° [31]. However, as shown in Figure 3, the phase difference increases very rapidly after 17.3 GHz, exceeding the phase margin and reaching 450° at 40 GHz. With FR-4 substrates, the phase unbalance levels can be significantly increased depending on the orientation between the DL line and the fiber weave, which may be very difficult, if not impossible, to estimate and compensate, especially with the mass PCB production. Also, in the presence of EM interference by adjacent DLs, the induced signal amplitude on each line of the DL can be different from each other.

Furthermore, common-mode noise induced by phase unbalance occupies a much wider frequency spectrum than that of the differential signal itself. Figure 4 shows the EM-simulated voltage spectral densities of a differential signal and common-mode noise in the presence of a phase unbalance. A non-return-to-zero (NRZ) pseudorandom bit sequence (PRBS) signal with a data rate of 40 Gbps is injected into the DL with a 60 mil length difference, monitoring the signal quality variations through EM simulations. As can be seen in Figure 4, the voltage spectral density of the differential signal is mostly distributed from near DC to 30 GHz, with a level of more than 10^−10^ V/GHz. On the other hand, the spectral density of the common-mode noise is distributed up to ~60 GHz. It is noted that the common-mode noise generated by the phase unbalance occupies a bandwidth almost two times wider than that of the differential signal. This necessitates suppressing it with a filter bandwidth at least twice wider than that of the differential signals to improve signal integrity. Therefore, undesirable phenomena, including the nonlinear phase interaction, the unbalanced EM interference, and the fiber weave effects of the FR4 substrates, limit the DL-based digital transmission lines to the maximum operation frequency of around 10 GHz, which is not suitable to meet the requirements of ultra-high-speed digital transmission for beyond 5G communications.

## 3. Properties of Balanced Lines (BLs)

The possibility of using balanced lines (BLs) as the ultra-high-speed digital data transmission lines with an excess of 40 GHz bandwidth was proposed by the authors’ group in [28]. The BL-based digital transmission structures are designed to be compatible with DL-based PCBs by utilizing the ultra-wideband DL-to-BL transitions. The practical planar BLs are a coplanar stripline (CPS) and a parallel stripline (PSL), as shown in Figure 5.

A CPS has two parallel lines on the same side of the substrate, and a PSL consists of two vertically parallel lines on the top and bottom of the substrate. The electric field lines of the BLs are mostly distributed between the signal lines, resulting in very tight EM coupling between the two signal lines. This tight EM coupling allows for maintaining the opposite polarity signals on each line and acts to restore the amplitude and phase unbalances when a distorted signal with phase and amplitude unbalances is introduced. To easily integrate the BLs into conventional DL-based digital circuits, a proper connection structure from the DL to the BLs should be adopted. A transition is a structure that provides a connection between different transmission lines, and its performance mostly determines the operation frequency bandwidth of the transmission lines. Previously, ultra-wideband DL-to-BL transitions were suggested by the authors’ group, providing frequency bandwidth from near DC to 40 GHz with a low insertion loss of less than 1.34 dB [28]. Therefore, the performances of the BLs can be maximized by using the proposed well-performing transitions in the DL-based digital circuit.

In utilizing the BLs for ultra-high-speed digital signal transmission, there are two very attractive properties: ultra-wideband common-mode rejection (amplitude balancing) and autonomous phase balancing. To validate the performance of the BLs, using 3D EM simulations, the *S*-parameter (*S_CC_*_21_; indicator for common-mode rejection) and the phase differences of the BLs are obtained as shown in Figure 6. The BLs are connected to the DL through the ultra-wideband transitions. In Figure 6a, it is observed that the common-mode rejection level of the CPS is greater than 7 dB from 13.2 GHz to 40 GHz, and that of the PSL is greater than 7 dB from 9.5 GHz to 40 GHz. Also, as shown in Figure 6b, the phase difference of the DL with a 30 mil length difference, after being compensated by an additional delay line, deviates more than 25° from 180° at frequencies above 22.7 GHz. On the other hand, for both the CPS and PSL, in the presence of a 30 mil length difference, the phase differences have deviations of less than 25° from 180° in frequency ranges from near DC up to 40 GHz. The results show that the BLs can reject the common-mode noise with a suppression level of 7~14 dB in the ultra-wideband frequency band and can restore the phase balance of the skewed differential signals without an extra compensation structure.

When only BLs are used in the presence of phase unbalances, providing tight EM coupling between the BLs, the highest frequency of the autonomous phase recovery can be limited to that corresponding to the length difference of around λ/8 (e.g., around 30 GHz with a 60 mil length difference of a DL on the 6 mil FR-4 substrate), and the common-mode suppression level is maintained at a level of 7~14 dB. In order to overcome this limitation, it is found, based on numerous observations of EM field configurations in the structures containing BLs, that additional conductor strips arranged laterally near the BLs can significantly enhance the levels of the common-mode rejection and extend the frequency bandwidth of autonomous phase recovery up to over 40 GHz, as described in the next section.

## 4. Common-Mode Rejection Structures Adjacent to Balanced Lines

Perspective views of the proposed common-mode rejection (CMR) structures are illustrated in Figure 7. A CPS-based CMR structure, presented in Figure 7a, consists of two uniplanar signal lines of the CPS and two arrays of linearly tapered conductor stubs, facing each other on the left and right sides of the CPS with a gap distance. Some areas of the tapered stubs are overlapped to form one conductor structure. On the other hand, a PSL-based CMR structure, as shown in Figure 7b, consists of two signal lines of the PSL and four arrays of linearly tapered conductor stubs on the top and bottom of the substrate.

### 4.1. Design of the CPS-Based CMR Structure

Top and bottom views of the proposed CPS-based CMR structure are presented in Figure 8. In the *AA′* to *BB′* section, a conventional DL is formed with a linewidth of *w_d_* and a gap distance of *g_d_* between the lines. In the section from *BB′* to *CC′*, the DL transforms into a CPS with a linewidth of *w_c_* and a gap distance of *g_c_*. In the *CC′* to *DD′* section, two CMR structures are facing each other on the left and right sides of the CPS with a gap distance of *g_rc_*. One of the CMR structures consists of an array of three linear-tapered conductor stubs with overlapped areas, where the total length is *l_bc_*, the width of the high-impedance part is *w_hc_*, the width of the low-impedance part is *w_lc_*, and the center-to-center gap distance of the tapered stubs is *g_bc_*. The CPS is transformed back to the DL. The dimensions of the design parameters are listed in Table 1.

### 4.2. Design of the PSL-Based CMR Structure

Figure 9 depicts top and bottom views of the proposed PSL-based CMR structure. In the *AA′* to *BB′* section, a conventional DL is formed with a linewidth of *w_d_* and a gap distance of *g_d_*. In the section from *BB′* to *CC′*, the DL transforms into a PSL with a linewidth of *w_p_*. In the section from *CC′* to *DD′*, on the top and bottom of the substrate, the two CMR structures are placed, facing each other on the left and right sides of the PSL, with a gap distance of *g_rp_*. One of the CMR structures consists of an array of three linear-tapered conductor stubs with overlapped areas, where the total length is *l_bp_*, the width of the high-impedance part is *w_hp_*, the width of the low-impedance part is *w_lp_*, and the center-to-center gap distance of the tapered stubs is *g_bp_*. The PSL is transformed back to the DL. The dimensions of the design parameters are listed in Table 2.

### 4.3. Simplified Electric Field Distributions of the Proposed CMR Structures

The CMR structures, placed close to the BLs, act to significantly drain the common-mode components on the BLs, while minimally affecting the differential signals. This action can be understood by observing the electric field distributions on the cross-sectional areas. Simplified electric field distributions at the cross-sections of the CPS-based and PSL-based CMR sections (*CC′* to *DD′* sections in Figure 8 and Figure 9) corresponding to the two signal modes (differential-mode or common-mode) are illustrated in Figure 10. With the differential signals carrying an opposite polarity signal at each signal line, the electric field lines are mostly distributed between the two signal lines of the CPS and PSL, respectively, as shown in Figure 10a,b. Only a negligible portion of the electric field lines links from the signal lines to the tapered CMR structures, and therefore the differential signals on the BLs, i.e., CPS and PSL, are not appreciably affected by the CMR structures.

On the other hand, with the common-mode signals, as shown in Figure 10c,d, most of the electric field lines link from the signal lines to the tapered CMR structures. Since the two signal lines have the same polarity, the electric field lines negligibly link between the signal lines of a BL (CPS or PSL), but tend to link from a signal line to the nearby conductor stub, therefore causing significant power flow of the common-mode components toward the conductor strip and resulting in significant common-mode rejection.

When a common-mode signal is applied to a BL consisting of two signal lines, the line impedance becomes very high, but indeterminate due to the absence of the nearby ground line or plane. Nevertheless, the common-mode signals manage to leak through the BL with an attenuation level of 7~14 dB. However, when the conductor stubs of various shapes are placed in close proximity to the BL, the common-mode signal easily finds a flow path through EM coupling to the conductor stubs, resulting in significant rejection of common-mode signals from the BLs. In this way, the tapered conductor stubs act to attain ultra-wide common-mode rejection bandwidth.

### 4.4. Fabrication and Measurements

The fabricated CPS- and PSL-based CMR structures are shown in Figure 11a,b, respectively. The two CMR structures are fabricated with the 8 mil Rogers 4003 substrate. The performances of the proposed CMR structures are measured with the 4-port vector network analyzer (VNA; Rohde & Schwarz ZNB40). In order to measure the input/output DLs for the 4-port *S*-parameters, a DL dividing structure, which splits a DL into two separate single-ended microstrip lines, is connected at both input and output ends. Figure 11c shows the measurement setup for the fabricated CMR structure, which is connected to the DL dividing structures. At the ends of the four separated microstrip lines, four end-launch connectors are installed (Southwest Microwave 1092-03A-5) for measurements with the VNA. The overall sizes of the CMR structures are listed in Table 3.

Since the additional auxiliary fixtures, which consist of the DL dividing structures and end-launch connectors, are connected to the proposed CMR structures, the contributions of these auxiliary fixtures should be removed from the overall measured results. The measured 4-port *S*-parameters are converted into the 2-port mixed-mode *S*-parameters through de-embedding techniques. Among various de-embedding techniques that extract the auxiliary effects from the raw-measured data, a 2X-Thru SFD (Smart Fixture De-embedding) method, which uses the data of two fixtures, is chosen. In [32], the *S*-parameters of the 1X-Fixture can be obtained from the 2X-Thru data by converting to the TDR (Time Domain Reflectometry) data, and the 1X-Fixture data are used to obtain the *S*-parameters of the DUT. The whole process of the 2X-Thru SFD is provided in AITT ver. 2024.05.15. (Advanced Interconnect Test Tool) software, which employs error correction techniques for accurate performance extraction [33]. Therefore, in this paper, AITT software is used to extract data from the measurements of the proposed CMR structures, removing auxiliary fixture effects.

Figure 12 compares the EM-simulated mixed-mode *S*-parameters with the de-embedded measured mixed-mode *S*-parameters of the proposed CMR structures. The insertion loss, return loss, and common-mode rejection level of the fabricated CPS-based CMR structures are presented in Figure 12a. The maximum insertion loss (*S_DD_*_21_; differential signal) of 3.95 dB is obtained from near DC to 40 GHz, and the return loss (*S_DD_*_11_) of the CMR structure is more than 10 dB from near DC to 40 GHz. Also, it is observed that the CMR structure rejects the common-mode signals (*S_CC_*_21_) from 6.17 GHz to 40 GHz with the suppression level of over 10 dB. In the case of the fabricated PSL-based CMR structure as shown in Figure 12b, the CMR structure provides the maximum insertion loss (*S_DD_*_21_) of 2.36 dB and the return loss (*S_DD_*_11_) of greater than 10 dB from near DC to 40 GHz. In addition, the common-mode rejection (*S_CC_*_21_) of the CMR structure is more than 10 dB from 4.92 GHz to 40 GHz. The measured insertion losses of the CMR structures are slightly higher than the EM-simulated results. Also, the measured return losses and common-mode rejection levels of the CMR structures show some deviations from the EM-simulated results, but are still greater than 10 dB. The deviations between the simulated and measured results may have occurred due to the fabrication tolerances and slight bending of the substrate by heat, moisture, or external pressure.

To figure out the causes of the dip in the common-mode rejection level (*S_CC_*_21_) as shown in Figure 12, especially at low frequency, isolines of the electric fields of the common-mode signal at the cross-section of the CPS-based CMR structure are depicted in Figure 13. The cross-section is cut in the lateral direction of the structure. As shown in Figure 13a, the resonant fields of the common-mode signal are formed around the structures in the frequency where a dip in the common-mode rejection level (*S_CC_*_21_) occurs, e.g., at 3.8 GHz. It is estimated that the half-wavelength of the common-mode signal on the proposed structure is matched with the lateral length of the structure, causing half-wavelength resonance. On the other hand, it can be observed that the common-mode signal at a different frequency (e.g., 10 GHz) is suppressed, as shown in Figure 13b. Therefore, the resonance at the specific frequency may enhance the common-mode signal transmission through the proposed structures; however, it is also noted that the common-mode signal is well suppressed at frequencies other than the resonance frequency.

## 5. Phase-Balancing Properties of the CMR Structures

### 5.1. Skew Compensation Designs

Figure 14a illustrates examples of length-matched DLs with curved corners in a conventional digital circuit, where the length compensation section is applied in a form of another bent section or an additional delay line section. In Section 2, it is shown that the length-matched DL has a limitation in the maximum operation frequency of around 10 GHz. On the other hand, it is demonstrated in Section 3 that the BLs have a capability of autonomous phase balancing for an incoming skewed signal, caused by the length difference of the DL, without an additional structure up to 20~30 GHz. In this section, it will be shown that the proposed CMR structures can extend the operation bandwidth of the autonomous phase recovery.

A skew test structure with a length compensation section for a DL is shown in Figure 13b, and a skew test structure for a CPS-based CMR structure is shown in Figure 14c. For both structures, the amount of signal skew is controlled by the length of a delay line on the left side of the circuit board. Both structures consist of DLs, having a linewidth *w_d_*, a gap distance *g_d_*, and a length of the delay line *l_d_*. Both the length compensation section (in Figure 14b) and the CMR structure (in Figure 14c) are connected at a distance of *l_s_*. The dimensions of the design parameters are shown in Table 4. Considering the relative permittivity of the substrate, the distance ls on the Rogers 4003 substrate (*ε_r_* = 3.55) should be less than 16.5 mm, following the guideline [30].

### 5.2. EM Simulation Results and Analysis

The phase differences of the differential signals for the length-matched DL, the BLs, and the BLs with the CMR structures are shown in Figure 14. The phase difference values are obtained from the differences between *S_SD_*_21_ and *S_SD_*_31_, which are the *S*-parameters of the test structure that contains a differential signal input and two single-ended signal outputs. A 60 mil length difference is chosen for the skew test by considering the maximum length-delayed situation in the digital circuit, such as line bending or interconnector layout.

As shown in Figure 15a, the phase difference of the length-matched DL deviates to over 25° from 180° after 18.3 GHz, and the difference increases as the frequency becomes higher. On the one hand, the CPS has a deviation of over 25° from 180° in the frequency band from 22.5 GHz to 34.4 GHz. When the length difference of the DL is large (i.e., in case of a deep signal skew), the CPS alone may have some limitations in recovering the skewed signal up to 40 GHz. On the other hand, when a CMR structure is placed in proximity to the CPS line, the skewed signal can be recovered with a deviation of less than 25° from 180° in the frequency band from near DC to 40 GHz.

A similar skew recovery test was performed with the PSLs. In Figure 15b, the phase difference of the length-matched DL is the same as that in Figure 14a. With the PSL alone, the phase difference has a deviation of over 25° from 180° in the frequency band from 28.1 GHz to 40 GHz. In this case, when a CMR structure is placed in proximity to the PSL, the skewed signal is recovered with a deviation of less than 25° from 180° in the frequency band from near DC to 40 GHz.

Therefore, both the CPS-based and PSL-based CMR structures can fully recover the phase balance of the skewed differential signal in the ultra-wideband frequency bandwidth up to 40 GHz, applicable for ultra-high-speed digital transmission.

To further observe the phase-balancing phenomenon of the CMR structure, the electric field distributions of the length-matched DL- and CPS-based CMR structures with the skewed differential signal at 25 GHz are shown in Figure 16. The red and blue colors indicate the positive and negative polarities of the differential signal, respectively. In Figure 16a, with the DL, a balanced differential signal is launched from the left side and becomes a skewed signal after passing a delay line. The unbalanced signal continues to propagate until it reaches the length compensation section on the right side. However, the compensation can be unsuccessful due to the nonlinear phase interactions between the two signal lines of the DL before the signal reaches the compensation section. The skewed amount after the compensation can be worse than before.

On the other hand, with the CPS-based CMR structure, as shown in Figure 16b, the skewed signal on the DL from the left side becomes more aligned in phase at the DL-to-CPS transition, and the phase recovery becomes enhanced as the signal passes through the CMR structure, obtaining a differential signal with better signal integrity. Therefore, it is demonstrated that the proposed CMR structures can enhance the capability of autonomous phase recovery, in addition to the improved common-mode rejection, up to over 40 GHz.

### 5.3. Fabrication and Measurements

The fabricated CPS-based CMR and PSL-based CMR structures to validate the phase-balancing property are shown in Figure 17a,b, respectively. To measure the *S*-parameters (*S_SD_*_21_ and *S_SD_*_31_) with the 4-port VNA, the DL dividing structures are connected to the structures.

The measured and EM-simulated phase differences of the skew test structures (length-matched DL-, CPS-, and CPS-based CMR structures) are shown in Figure 18a. Since the de-embedding method is not applicable to the skew tests, which rely on 3-port measurements, the measured results include the effects of a DL dividing structure and end-launch connectors. This may have contributed to the discrepancies between the simulated and measured data. It is observed that the phase difference of the length-matched DL has a deviation of less than 25° from 180° in the frequency band up to 17.5 GHz. On the other hand, the phase difference of the CPS is less than 25° from 180° from near DC to 22.6 GHz, and the fabricated CPS-based CMR structure has a phase difference of less than 25° from 180° from near DC to 40 GHz. For the PSL and PSL-based CMR structures, as shown in Figure 18b, the phase difference of the PSL is less than 25° from 180° from near DC to 28.4 GHz, and the fabricated PSL-based CMR structure provides a phase difference of less than 25° from 180° from near DC to 36 GHz. In the frequency band from 36 GHz to 40 GHz, the maximum phase difference of the proposed CMR structure is 212.7°. Therefore, despite some deviations between the simulated and measured results, it is demonstrated that the proposed CMR structures have enhanced phase-balancing properties, which can recover the phase balance of the skewed signal.

In Figure 18, the DL-to-BL transition structure is applied at a distance (*l_s_*) of 600 mil from the location of the signal skew generation, and therefore the amount of nonlinear phase interaction becomes significant and the signal balance is severely distorted, as shown in the length-matched DL case in Figure 18a,b. If the BL or the BL with the proposed CMR structure is applied at a closer distance from the signal skew source, the phase recovery will be significantly improved. It is noted that the proposed DL-to-BL transition structure does not occupy a wider area than the DL layout.

In addition, in order to demonstrate the common-mode rejection level of the proposed CMR structures, the common-mode voltages after applying the phase compensation structures are monitored. In Figure 19, common-mode voltages are shown when a long stream of a PRBS, which has a rise time of 2.5 ps, a unit interval of 25 ps, a pulse voltage of 1.0 V, and a bit length of 800 bits, is excited to the skew test structures (length-matched DL-, CPS-, and PSL-based CMR structures). It can be observed that the length-matched DL structure has a maximum peak CM voltage of 0.12 V. On the other hand, the CPS-based CMR and the PSL-based CMR structures have maximum peaks at the CM voltages of 0.03 V and 0.04 V, respectively. Therefore, the proposed CMR structures are again confirmed to have ultra-wideband common-mode rejection performance.

The performances of the proposed CMR structures are compared with those of the reported ultra-wideband CMR devices in Table 5. The fractional bandwidths of the reported CMR devices were less than 138%. Also, the CMR bandwidths of the reported devices covered frequencies less than 20 GHz. On the other hand, the proposed CMR structures provide a wider fractional bandwidth of 146.5% and 156.2%. Also, the proposed structures can reject the common-mode signal at frequencies of around 40 GHz. Furthermore, only the suggested structures have the unique phase-balancing property by themselves without employing external circuitries.

Consequently, based on the measured and EM-simulated results, the proposed CMR structures utilizing BLs demonstrate the ability to provide simultaneously enhanced capabilities of common-mode rejection and phase balancing in the frequency bandwidth of several 10s of GHz for ultra-high-speed digital transmission. In addition, the proposed CMR structures can be applied to the DL-based digital circuit using ultra-wideband DL-to-BL transitions, supporting better signal integrity of the ultra-high-speed digital circuit.

## 6. Conclusions

In this paper, ultra-wideband digital transmission structures with enhanced common-mode rejection (CMR) and autonomous phase-balancing capabilities up to over 40 GHz are proposed for ultra-high-speed digital transmission, required by 5G/6G communications. The common-mode noise, mostly caused by unbalances in amplitude and/or phase, is a main source of EM interference and RFI, which degrade signal integrity and damage the digital circuits.

Up to now, a differential line (DL) has typically been used as a digital transmission line for high-speed digital signals. However, due to nonlinear phase interactions between the two signal lines of the DL and unbalanced EM interferences by adjacent lines, the common-mode noise inevitably occurs in a complex digital circuit, limiting the maximum operating frequency at around 10 GHz. On the other hand, the utilization of a balanced line (BL) as a digital transmission line provides numerous advantages that can help overcome the limitations of DLs, by providing ultra-wideband common-mode rejection (7~14 dB) and autonomous phase balancing of the distorted digital signal. This autonomous phase recovery action using only BLs, however, can be limited by the phase unbalance level of the incoming signal within the phase unbalance of ~λ/8.

To significantly enhance the common-mode rejection capability as well as the phase recovery action, BL-based common-mode rejection structures are proposed, by laterally arranging extra conductor strips adjacent to the BLs. It is shown that the implemented CPS-based and PSL-based CMR structures have a common-mode rejection level of more than 10 dB in the frequency bandwidth from 6.17 GHz to 40 GHz and from 4.92 GHz to 40 GHz, respectively. Also, it is observed that the implemented CMR structures can recover the phase balance of a skewed signal from near DC up to over 40 GHz. Through the EM simulations and measurements, the proposed ultra-wideband digital transmission structures are demonstrated to perform in ultra-wideband bandwidth with excellent phase recovery and common-mode rejection.

The proposed structures can be easily fabricated on typical PCBs with only a slight cost increase through commercial PCB manufactures. Therefore, the proposed structures can readily be applied to the existing high-speed digital circuit boards and will be able to improve performance on a variety of high-speed interfaces, such as USB and PCIe. Further, with fully utilizing these proposed structures, digital circuit boards can be implemented, supporting ultra-high-speed digital transmission in excess of ~100 Gbps required by 5G/6G communications.

## Figures and Tables

**Figure 1 sensors-24-06180-f001:**
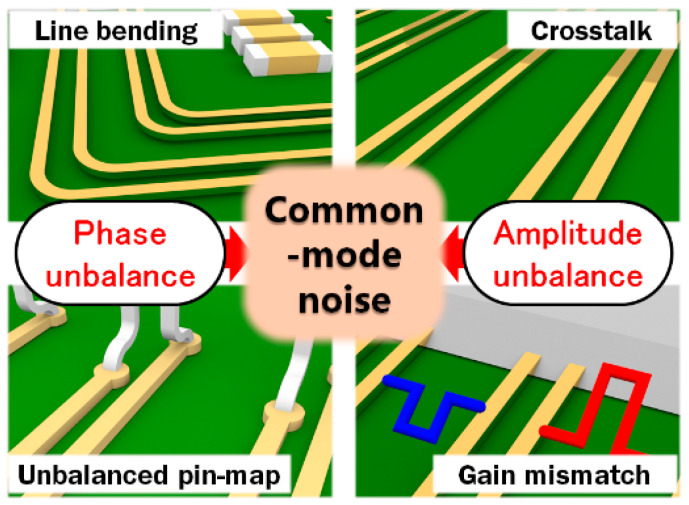
Various cases of common-mode noise generation in a digital circuit environment.

**Figure 2 sensors-24-06180-f002:**
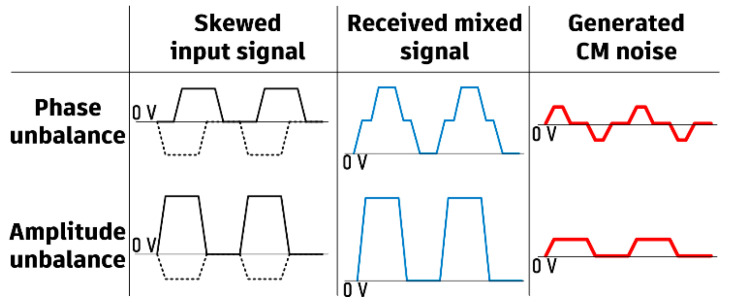
Waveforms of the two signal lines of a DL with a phase unbalance and/or an amplitude unbalance, causing common-mode noise.

**Figure 3 sensors-24-06180-f003:**
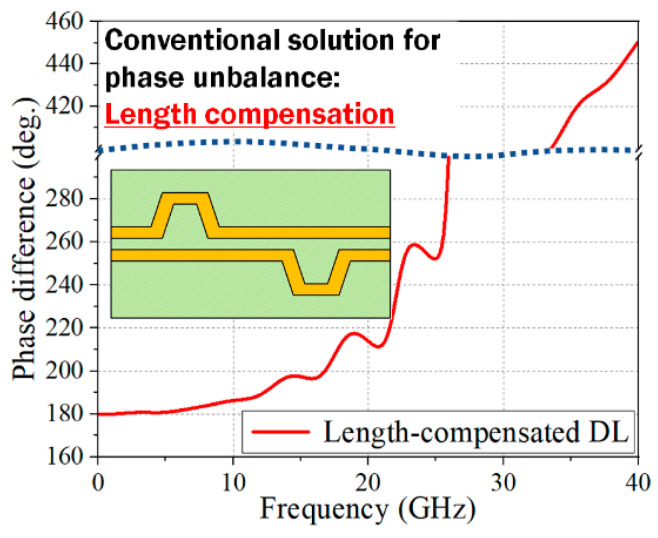
EM-simulated phase difference between two differential signal lines with a length compensation method. The line difference between the two DLs is 60 mil.

**Figure 4 sensors-24-06180-f004:**
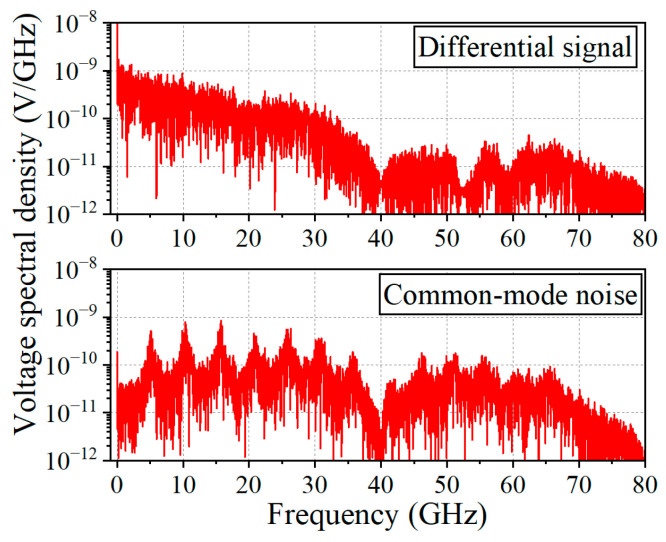
EM-simulated voltage spectral densities of a differential signal and common-mode noise. A random digital signal with data rate of 40 Gbps is injected into the DL structure, and the spectrum is monitored after applying the compensation structure (a 60 mil-delayed DL).

**Figure 5 sensors-24-06180-f005:**
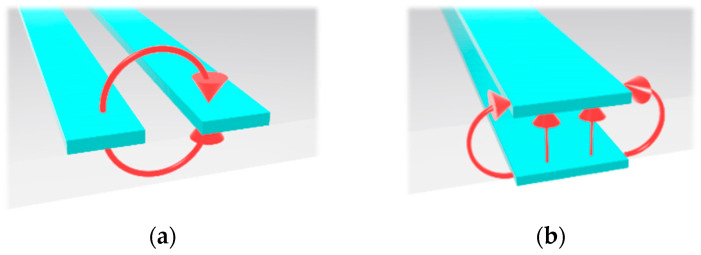
Perspective views of the balanced lines with simplified electric field distributions: (**a**) coplanar stripline (CPS) and (**b**) parallel stripline (PSL).

**Figure 6 sensors-24-06180-f006:**
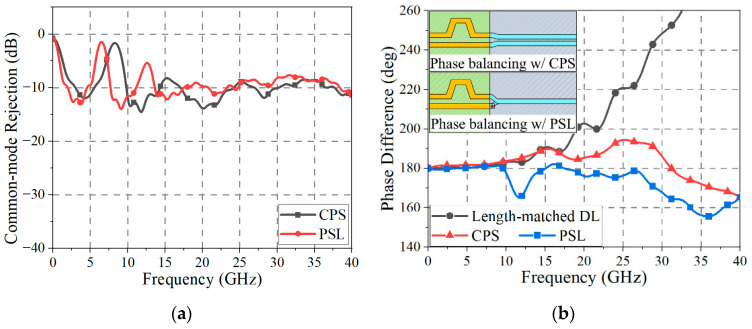
EM-simulated performances of the BLs: (**a**) ultra-wideband common-mode rejection and (**b**) autonomous phase balancing when a length difference of 30 mil occurs.

**Figure 7 sensors-24-06180-f007:**
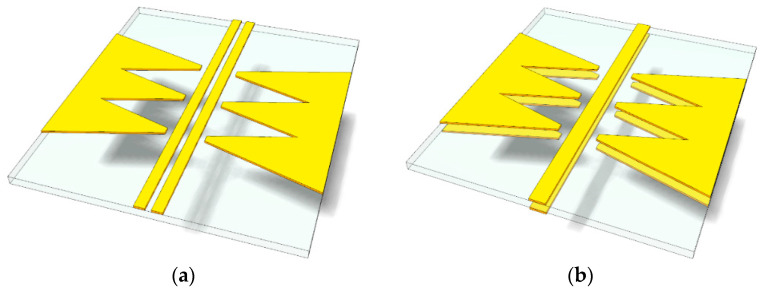
Perspective views of the proposed common-mode rejection (CMR) structures: (**a**) CPS-based CMR structure and (**b**) PSL-based CMR structure.

**Figure 8 sensors-24-06180-f008:**
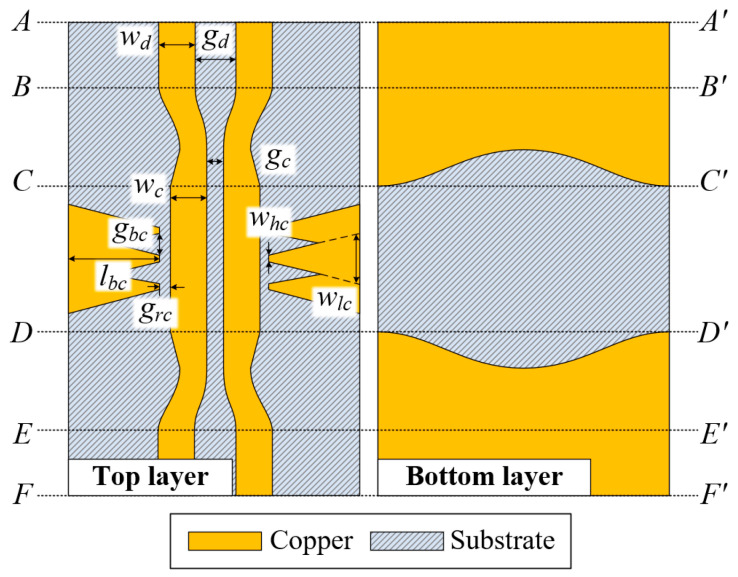
Top and bottom views of the proposed CPS-based CMR structure.

**Figure 9 sensors-24-06180-f009:**
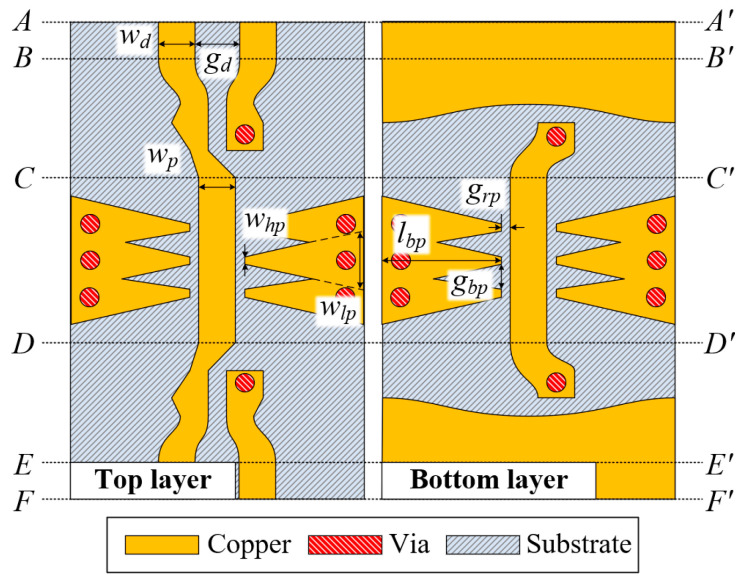
Top and bottom views of the proposed PSL-based CMR structure.

**Figure 10 sensors-24-06180-f010:**
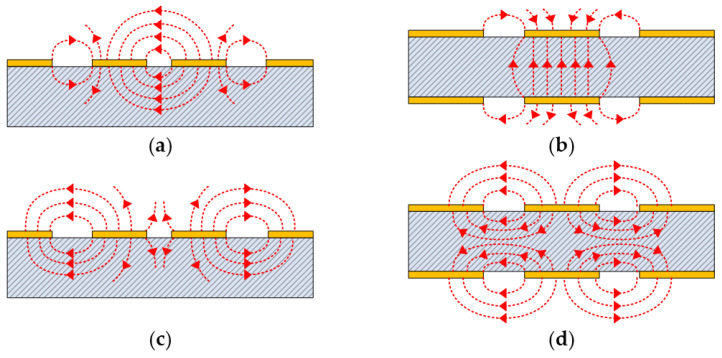
Simplified electric field distributions at the cross-sections of the BLs placed with adjacent tapered conductor strips: (**a**) CPS with a differential signal, (**b**) PSL with a differential signal, (**c**) CPS with a common-mode signal, and (**d**) PSL with a common-mode signal.

**Figure 11 sensors-24-06180-f011:**
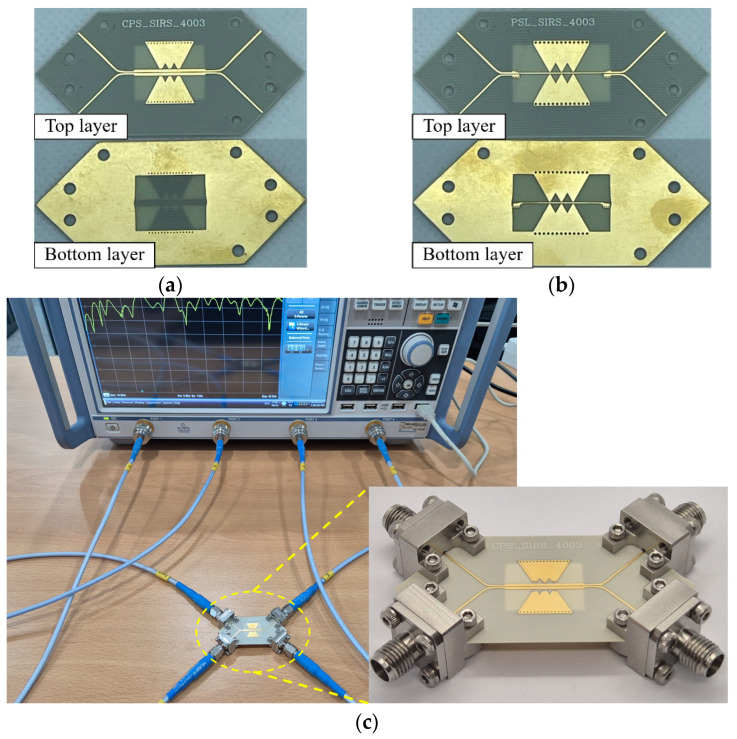
Fabricated CMR structures with the DL dividing structures: (**a**) CPS-based CMR structure, (**b**) PSL-based CMR structure, and (**c**) picture of device under test (DUT) (i.e., CPS-based CMR structure) measurement setup.

**Figure 12 sensors-24-06180-f012:**
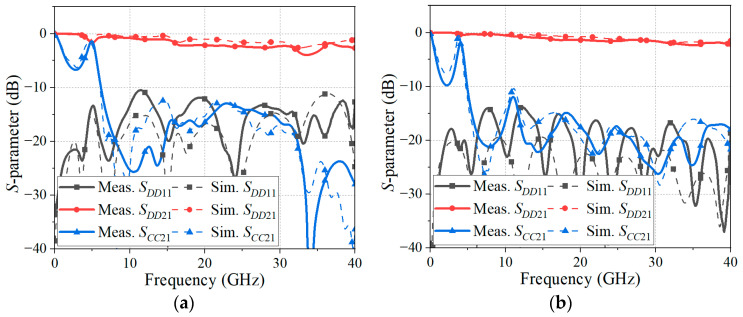
EM-simulated and measured *S*-parameters of the proposed CMR structures: (**a**) CPS-based CMR structure and (**b**) PSL-based CMR structure.

**Figure 13 sensors-24-06180-f013:**
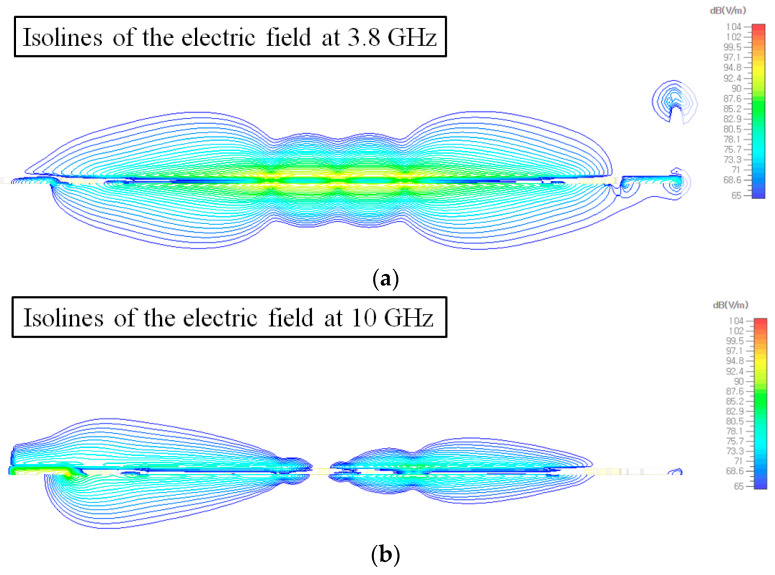
Isolines of the electric fields at the cross-section of the CPS-based CMR structure at (**a**) 3.8 GHz and (**b**) 10 GHz. The cross-section is cut in the lateral direction. The common-mode signal is transmitted from the left side of the structure to the right side. The amplitude of the electric field is colored from the blue color for the minimum level to the red color for the maximum level.

**Figure 14 sensors-24-06180-f014:**
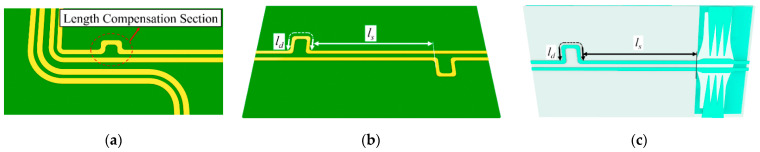
(**a**) Examples of the length-matched DLs with curved corners, (**b**) skew test structure of the length-matched DL, and (**c**) skew test structure of the CPS-based CMR structure.

**Figure 15 sensors-24-06180-f015:**
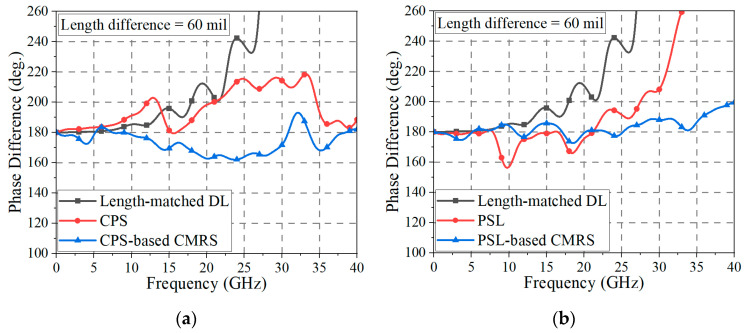
Phase difference of the length-matched DL is compared with (**a**) CPS and CPS-based CMR structure and (**b**) PSL and PSL-based CMR structures. The results are EM-simulated with *l_d_* = 60 mil.

**Figure 16 sensors-24-06180-f016:**
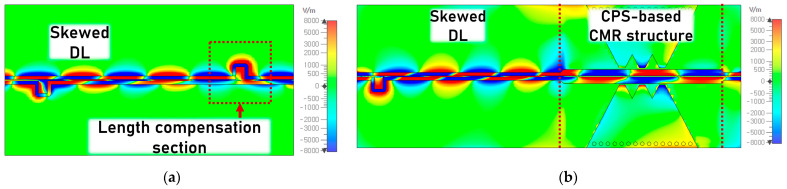
Top view of the electric field distributions of the skew test structures: (**a**) length-matched DL- and (**b**) CPS-based CMR structures. The positive and negative polarities of the electric field are illustrated with the red and blue colors, respectively. Also, the amplitude of the electric field is represented by the gradual changes in the colors (i.e., the zero level is expressed as a green color, and the maximum level is a red or blue color).

**Figure 17 sensors-24-06180-f017:**
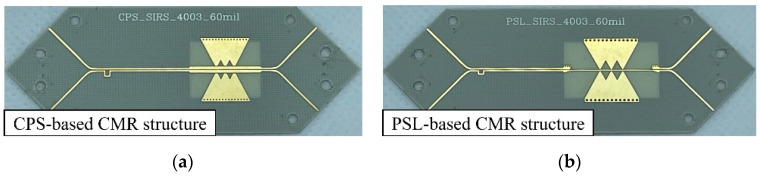
Fabricated skew test structure of the proposed CMR structures: (**a**) CPS-based CMR structure and (**b**) PSL-based CMR structure.

**Figure 18 sensors-24-06180-f018:**
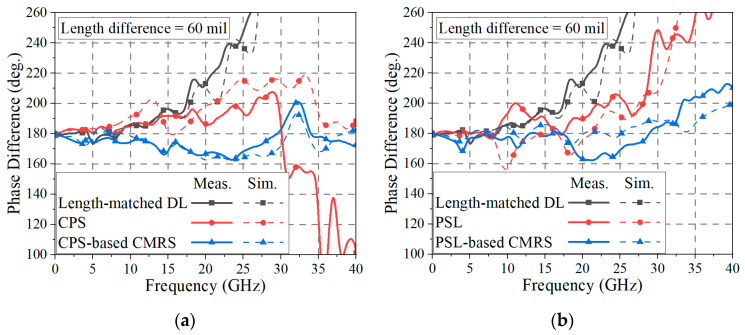
Phase difference of the length-matched DL is compared with (**a**) CPS and CPS-based CMR structure and (**b**) PSL and PSL-based CMR structures. The results are measured and EM-simulated with *l_d_* = 60 mil.

**Figure 19 sensors-24-06180-f019:**
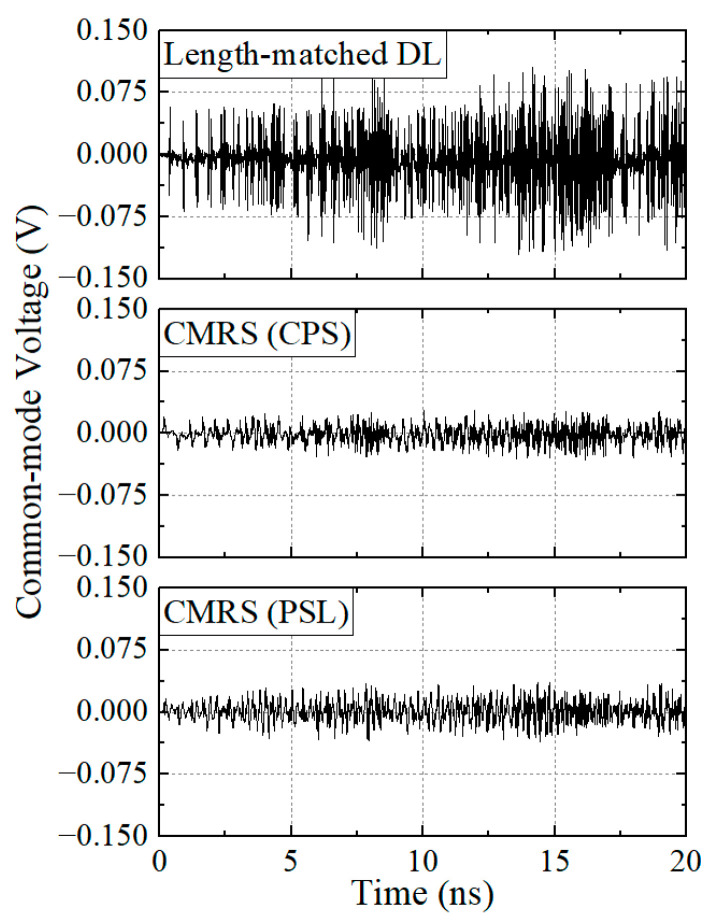
Common-mode voltages of the skew test structure of the length-matched DL-based CMR structure, the CPS-based CMR structure, and the PSL-based CMR structure.

**Table 1 sensors-24-06180-t001:** Design parameters of the proposed CPS-based CMR structure.

**Parameters**	*w_d_*	*g_d_*	*w_c_*	*g_c_*	*g_bc_*	*g_rc_*	*l_bc_*	*w_hc_*	*w_lc_*
**Size in mil** **(mm)**	11.36 (0.29)	5 (0.13)	22.29 (0.57)	5 (0.13)	50 (1.27)	5 (0.13)	220 (5.59)	20 (0.51)	250 (6.35)

**Table 2 sensors-24-06180-t002:** Design parameters of the proposed PSL-based CMR structure.

**Parameters**	*w_d_*	*g_d_*	*w_p_*	*g_bp_*	*g_rp_*	*l_bp_*	*w_hp_*	*w_lp_*
**Size in mil** **(mm)**	11.36 (0.29)	5 (0.13)	7.5 (0.19)	82 (2.08)	5 (0.13)	216 (5.49)	8 (0.20)	250 (6.35)

**Table 3 sensors-24-06180-t003:** Overall sizes of the fabricated CMR structure.

Structure	Size [mil (mm)] *
CPS-based CMR structure	570 × 500 × 8 (14.48 × 12.70 × 0.20)
PSL-based CMR structure	770 × 500 × 8 (19.56 × 12.70 × 0.20)

* Structure dimensions: (length in transverse axis) × (length in longitudinal axis) × (height of the structure).

**Table 4 sensors-24-06180-t004:** Design parameters of the simulation structures.

**Parameters**	*w_d_*	*g_d_*	*l_s_*	*l_d_*
**Size in mil** **(mm)**	11.36 (0.29)	5 (0.13)	600 (15.24)	60 (1.52)

**Table 5 sensors-24-06180-t005:** Compared performances of the proposed CMR structure with the reported CMR devices.

Ref.	Method	Bandwidth [GHz] (FBW)	Suppression Level [dB]	Insertion Loss [dB]	Size [mm × mm]	Phase-Balancing Property
[21]	DGS	3.7–10.8 (98%)	10	3.0	6.63 × 6.63	X
[22]	DGS	3.2–12.4 (118%)	15	N/A	10 × 10	X
[23]	PCB-embedded	4.6–20 (125%)	7	7	10.2 × 2.2	X
[24]	Absorptive CM filter	2–11 (138%)	10	5	15 × 9.6	X
[25]	EBG	3.1–17 (138%)	10	N/A	47.8 × 17.3	X
[26]	Quarter-wavelength resonator	3.53–10.1 (96%)	10	4.48	N/A	X
This work (CPS-based)	BLs with conductor strips	6.17–40 (146.5%)	10	3.95	14.48 × 12.70	O
This work (PSL-based)	BLs with conductor strips	4.92–40 (156.2%)	10	2.36	19.56 × 12.70	O

## Data Availability

Data are contained within the article.

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
