# Peer review of "Ultra-Wideband Common-Mode Rejection Structure with Autonomous Phase Balancing for Ultra-High-Speed Digital Transmission"

_sensors, 2024, doi:10.3390/s24196180_

Round 1

Reviewer 1 Report

Comments and Suggestions for Authors

The paper is well written, especially the introduction and simulation part, but the fabrication and measurement part needs some improvements before publication.

1. Show the complete experimental setup, including the VNA, end-launch connectors, and connected fabricated structures.

2. What are the challenges in fabrication, and how can you minimize the differences between simulation and measurements? 

3. Add a comparison table with the state-of-the-art devices.

Comments on the Quality of English Language

Minor editing of English language required.

Reviewer 2 Report

Comments and Suggestions for Authors

The wideband digital transmission structures are proposed. These structures can restore the phase and amplitude of a skewed digital signal, due to that the common-mode rejection level and the autonomous phase balance up to over 40 GHz is realized. The laterally arranging extra conductor strips adjacent to the balance line are proposed, therefore, the common-mode rejection capability enhanced. Besides the simulations, the model structures are fabricated and measured. The results of the simulations are mostly validated by the measurements. The proposed ultra-wideband digital transmission up to over 40 GHz are proved by the EM-simulations and measurements. So, the structures proposed demonstrate ability to operate in ultra-wideband bandwidth with good phase recovery and common-mode rejection. The paper falls into the scope of the journal. Despite the volume, Introduction gives a useful review in the considered field. The results of the paper satisfy the requirements of 5G/6G communications and possibly will be applied in the ultra-high-speed digital data transmission.

The paper needs only minor corrections before publication.

In caption to Figure 15 it is worthy to mention, which component of the electric field is shown.

Many times duplicating explanations of the abbreviations encounter. There are, for example, (BLs) line 225, (CPS) and (PSL) line 229, (CMR) line 280, (DL) line 291, (DL) line 303 and others.  The duplicating explanations present even in Conclusions: (CMR) line 513, (DL) line 518, (BL) line 523. In addition to that, the abbreviation PRBS in line 497 is introduced but never used in the text below.

Reviewer 3 Report

Comments and Suggestions for Authors

The authors have designed an ultra-wideband common mode suppression structure with automatic phase balance for ultra-high speed digital transmission. The designed mechanism has certain engineering application value and can provide reference for related researchers. The shortcomings of the paper are as follows:

1. The main contributions of the paper need to be listed in sections.

2. The paper and the latest achievements should be fully compared and analyzed.

3. If possible, it is best for the authors to verify the actual effect of specific application scenarios.

Comments on the Quality of English Language

This paper is easy to read and understand.

Reviewer 4 Report

Comments and Suggestions for Authors

Ultra-wideband digital transmission structures with enhanced common-mode rejection (CMR) and autonomous phase balancing capabilities up to over 40 GHz are proposed for ultra-high-speed digital transmission, required by 5G/6G communications. The manuscript is well prepared and well written. The specific comments are listed as follows:

1.      It seems that the proposed CPS-based and PSL-based CMR structures are much more complex than the conventional solution, apart from the given transmission performance, it is suggested to make a thorough comparison, including the bandwidth, return loss, insertion loss, size, complexity, cost, etc.

2.      Why there is a spur for Scc21 at around 5 GHz?

3.      Why the simulated and measured results show great deviation?
